# New Horizons of Macrophage Immunomodulation in the Healing of Diabetic Foot Ulcers

**DOI:** 10.3390/pharmaceutics14102065

**Published:** 2022-09-27

**Authors:** Ching-Wen Lin, Chien-Min Hung, Wan-Jiun Chen, Jui-Ching Chen, Wen-Yen Huang, Chia-Sing Lu, Ming-Liang Kuo, Shyi-Gen Chen

**Affiliations:** 1Oneness Biotech Co., Ltd., Taipei 106, Taiwan; 2Microbio Co., Ltd., Taipei 115, Taiwan; 3Division of Plastic and Reconstructive Surgery, Department of Surgery, Tri-Service General Hospital, National Defense Medical Center, Taipei 114, Taiwan

**Keywords:** diabetic foot ulcer, wound healing, M1/M2 macrophage, exosome, microRNA, adipocyte-derived stem cell, reactive oxygen species (ROS), *Plectranthus amboinicus*

## Abstract

Diabetic foot ulcers (DFUs) are one of the most costly and troublesome complications of diabetes mellitus. The wound chronicity of DFUs remains the main challenge in the current and future treatment of this condition. Persistent inflammation results in chronic wounds characterized by dysregulation of immune cells, such as M1 macrophages, and impairs the polarization of M2 macrophages and the subsequent healing process of DFUs. The interactive regulation of M1 and M2 macrophages during DFU healing is critical and seems manageable. This review details how cytokines and signalling pathways are co-ordinately regulated to control the functions of M1 and M2 macrophages in normal wound repair. DFUs are defective in the M1-to-M2 transition, which halts the whole wound-healing machinery. Many pre-clinical and clinical innovative approaches, including the application of topical insulin, CCL chemokines, micro RNAs, stem cells, stem-cell-derived exosomes, skin substitutes, antioxidants, and the most recent Phase III-approved ON101 topical cream, have been shown to modulate the activity of M1 and M2 macrophages in DFUs. ON101, the newest clinically approved product in this setting, is designed specifically to down-regulate M1 macrophages and further modulate the wound microenvironment to favour M2 emergence and expansion. Finally, the recent evolution of macrophage modulation therapies and techniques will improve the effectiveness of the treatment of diverse DFUs.

## 1. Introduction

Diabetes is a chronic metabolic disease characterized by an elevated level of blood glucose, which eventually leads to serious complications, including cardiovascular disease, diabetic nephropathy, neuropathy and retinopathy, peripheral artery disease, and foot ulceration. The prevalence of DFUs is approximately 6.3% globally [1], and the lifetime incidence of foot ulcers is estimated to be 19–34% among persons with diabetes worldwide [2]. DFUs are a primary reason for hospitalization and amputation among patients with diabetes, thus contributing to a huge medical burden globally; moreover, the 5-year mortality rate for patients with DFUs is 42% [3,4]. Furthermore, a 10-year cohort study reported that the incidences of contralateral amputation and re-amputation per 100 amputee-years in female amputees with diabetes were 15 and 16, respectively, whereas in male amputees with diabetes, these figures were 18 and 21, respectively [5]. One in four amputees may require contralateral amputation and/or re-amputation [5]. The global disability burden attributed to DFUs is considerable and was estimated to be 2.5 million years lived with disability in 2016 [6]. A 15-year follow-up cohort study performed in Taiwan found that the event-year medical cost of patients with DFUs was USD 4701.95 [7]. In addition to the direct cost of the treatment of foot ulcers, there are other indirect expenditures that possibly contribute to loss of productivity, family costs, family status, and loss of quality of life.

To resolve the problem of DFUs, the execution of the standard of wound care, including debridement, off-loading, infection control, and the maintenance of a moist environment using dressings, is recommended [3,8]; in contrast, adjunctive therapies, such as the use of hyperbaric oxygen, and negative pressure wound therapies are applied if the DFUs worsen [9]. Advanced therapies, such as the administration of recombinant human platelet-derived growth factor subunit B (PDGF-BB) and tissue engineering products, which directly provide the growth factors required for late-stage wound healing, still present some limitations and the annual amputation rate is on the rise [10,11]. Therefore, alternative approaches that benefit patients with DFUs are still needed.

The monocyte–macrophage lineage is characterized by diversity and plasticity in response to the local environment, which is critical for host defence and innate immunity. Current studies indicate that macrophages are more complex as they have different subtypes that participate in multiple physiological functions and disease pathologies [12]. Macrophages can be divided into two predominant types: the classical M1 lineage, which displays pro-inflammatory functions, and the alternative M2 lineage, which exhibits an anti-inflammatory phenotype [13]. Both M1 (pro-inflammatory) and M2 (pro-healing) macrophages are required for the progression of the wound-healing process [13,14,15]. Thus, recent research has focused on the pathological role of macrophages in diabetic wound healing and efforts have been dedicated to the reshaping of dysregulated macrophages using different approaches. Examples of emergent approaches aimed at modulating the macrophage subtypes, especially the down-regulation of M1 macrophages and the enrichment of M2 macrophages, will be discussed in this review. Of note, numerous innovative biomaterials were created to provide suitable environments for recruiting or polarizing the distinct types of macrophages [16,17] that have been implicated in inflammation-associated conditions and tissue regeneration (as reviewed in [18,19,20]).

## 2. Role of M1 and M2 Macrophages in Wound Healing

Immediately after the homeostasis stage, M1 macrophages appear at the site of injury and act in pathogen phagocytosis by destroying/removing damaged cells during the inflammation stage. M1 macrophages also activate other immune cells through the release of pro-inflammatory cytokines (TNF-α, IL-1β, and IL-6) and chemokines (CXCL8-11), which exert positive feedback on inactivated monocytes or antigen-presenting cells (APCs) [12,21]. M1 macrophages can be characterized as promoting the positive expression of co-stimulatory molecules (CD86 and CD80), toll-like receptor-2 (TLR-2), TLR-4, and MHC-II on the cell surface, as well as increasing the activities of nitric oxide (NOS) and reactive oxygen species (ROS) [12,22]. The transition of the healing process to the proliferative/regenerative stage triggers a dynamic switch of the macrophage population from the M1 to the M2 type [12,22,23].

M2 macrophages function as natural feedback regulators from M1 macrophages and can be further classified into four subtypes, i.e., M2a, M2b, M2c, and M2d, according to their biological functions and phenotypes (secreted cytokines and surface markers) [21,24,25]. The M2a subtype is characterized by high levels of CD206, FIZZ1, TGF-β, CCL17, and CCL22 expression induced by IL-4 and IL-13, which function as anti-inflammatory and wound-healing factors [24,26]. The M2c macrophages express CD206, CD163, and MERTK; produce elevated levels of TGF-β, IL-10, MMP-9, and IL-1β; and can be stimulated by IL-10 and glucocorticoids [22,27]. Both M2a and M2c subtypes are considered pro-healing and pro-remodelling macrophages [26], which induce the migration and proliferation of fibroblast and keratinocytes, as well as recruit endothelial stem cells to allow the development of granulation tissue and neovascularization. Secretion of cytokines, chemokines, and growth factors is critical for co-ordinating different cells in the proliferation/remodelling stage of normal wound healing [28,29]. Such cross-talk between different cells was impaired in diabetic wounds [22].

In addition, M1 macrophages are essential for the initiation of sprouting angiogenesis through stimulating the secretion of proangiogenic cytokines and VEGF, while M2 macrophages are required for subsequent vessel regression and remodelling [30]. M2 macrophages also produce matrix metalloproteinase (MMP) and VEGFs, which are essential for remodelling [31]. Thus, the proper switch from M1 to M2 macrophages at the end of the pro-inflammation stage to the proliferation/remodelling stage, which allows for sufficient levels of M2 macrophages, is a key step for proceeding to normal wound healing.

## 3. Switch of M1 and M2 Macrophages at the Molecular Level

M1 and M2 macrophages can be polarized from monocyte/resting macrophages locally, around the wound, or can be recruited from peripheral areas [15,32]. In the wound-healing process, M2 and M2 subtype macrophages can be produced by switching from M1 macrophages; this type of switch could be driven by environmental changes in cytokines, miRNAs, transcription factors, and exosomes [15,31,32,33,34]. The proper attenuation of innate immune or pro-inflammatory cytokines, such as IFN-γ, TNF-α, and IL-1β, which are important for M1 polarization and survival, enhances the expression of M2-associated markers (CD206, CD36, or CD163) and cytokines (TGF-β, CCL2, CXCL8, and CCL3) [35,36]. These extra-cellular factors then trigger the M2-polarizing signalling and switch the molecular hub from an M1-associated profile (NF-κB-, IRF5-, STAT1-, and AKT2-dominant) to an M2-associated (IRF4-, STAT6-, PPAR-γ-, and AKT1-dominant) trend [32]. In addition, IL-10, which is an anti-inflammatory cytokine that is important for the regeneration/remodelling phase of wounds, has been shown to increase the amount of the IL-4R chain on the cell surface, thus rendering macrophages more sensitive to IL-4 and IL-13 and triggering an M2-polarizing trend [37]. The IL-4/IL-13-mediated signalling pathway further drives IL-10R, IL-6R, and IL-4R through STAT3 [32,37]. This feedback regulation between anti-inflammatory cytokines reshapes the environment to favour M2 polarization.

## 4. Dysregulation of the M1/M2 Switch in DFUs

Diabetic wounds feature the persistence of chronic inflammation and a delayed proliferation/remodelling phase characterized by a stalled non-healing state, in which the dysregulation of innate immune cells results in high neutrophil and M1 macrophage counts; in addition, defects in M2 polarization are observed and cause a prolonged inflammation phase [15,38,39]. Several factors disturb the healing process of diabetic wounds, including hyperglycaemia, advanced glycation end-products, and impaired angiogenesis [39,40,41]. These factors result in a disordered immune spectrum, such as excessive pro-inflammatory cytokines and immune cells (neutrophils and M1 macrophages) accompanied by a defect in the M1-to-M2 switch [39,42,43]. Specifically, a lower number of M2 macrophages and a higher M1/M2 ratio result in the shortage of the EGF, FGF, PDGF, and VEGF growth factors, as well as of the IL-10, TGF-α, and TGF-β anti-inflammatory cytokines, which are all crucial factors for the proliferation and remodelling stages [22,44]. Thus, recent studies have emphasized M2-enrichment approaches, such as the administration of M2 macrophages directly or the elimination of factors that obstruct M2 recruitment or polarization, for dealing with DFUs.

## 5. Innovative Therapies for Diabetic Foot Ulcers

### 5.1. Topical Insulin

Non-healing diabetic wounds are typically attributed to long-standing complications in metabolism triggered by hyperglycaemia. Briefly, elevated blood glucose stimulates the secretion of pro-inflammatory cytokines, such as IL-6, IL-1β, and TNF-α, from macrophages, thus encouraging a vicious cycle of persisting M1 macrophage phenotypes [44]. Systemic insulin administration is an effective way to improve blood glucose control; in turn, the effects of locally delivered insulin on diabetic ulcers have been investigated. Mechanistically, insulin was proven to play a critical role in the switching of M1 to M2 macrophage polarization through the phosphatidylinositol 3-kinase (PI3K)/AKT pathway and peroxisome proliferator-activated receptor-gamma (PPAR-γ) signalling in vitro and in STZ-induced diabetic rats that received insulin subcutaneously [45]. The effect was also sustained when topical insulin was applied to diabetic wounds in rats, in which insulin promoted the efferocytosis of apoptotic neutrophils by macrophages, thus expediting macrophage polarization from the M1 to the M2 type [46]. Interestingly, excessive levels of insulin-degrading enzymes were observed in diabetic wound beds, explaining why local insulin deficiency may occur even when systemic insulin is elevated [46]. Recently, topical insulin therapeutics in various formulations, including local injection, spray, cream, dressing, and liposome-containing gel, were evaluated in clinical studies [47,48]. In particular, an insulin-loaded liposomal chitosan gel provided sustained insulin release over 24 h [49]. In a recent trial, insulin-containing dressings decreased ulcer sizes by 49.7% (*n* = 55) compared with a reduction of only 19.2% in the control group (which received saline dressings; *n* = 55) after a 2-week treatment (*p* = 0.001) [50]. Other clinical studies also demonstrated that topical insulin administration accelerated wound healing without causing hypoglycaemia [51,52]. Nevertheless, the high-quality evidence necessary to fully evaluate the clinical efficacy of topical insulin therapy in DFUs remains limited.

### 5.2. Chemokines

Numerous cytokines and chemokines are involved in the normal wound-healing process. One such chemokine, CCL2 (MCP-1), is markedly induced in injured skin in healthy individuals, where it serves as a potent macrophage chemoattractant [53]. However, diabetic wounds exhibited a significant delay in the macrophage response and attenuated CCL2 expression early after injury in mice [53,54]. Topical administration of CCL2 immediately after injury restored normal wound closure, as evidenced by enhanced neovascularization and collagen accumulation. The pro-healing function of CCL2 was attributed to the recruitment of macrophages accompanied by enhanced VEGF and TGF-β secretion [53]. Interestingly, a recent study reported that the epidermal-keratinocyte-derived CCL2 induced macrophages to secrete EGF, which activates basal epidermal keratinocyte proliferation in db/db mice [27]. Thus, CCL2 supplementation appears to be an effective therapeutic option for the re-initiation of normal inflammation and subsequent proliferation and for the remodelling of non-healing diabetic ulcers. Because of promising results in animals, clinical studies on the effect of topical CCL2 application on DFUs are urgently needed.

### 5.3. Advanced Glycosylation End Products (AGEs)

Advanced glycosylation end products (AGEs) are formed via the nonenzymatic glycation of proteins upon exposure to reducing sugars, which are also called “glycotoxins” [55]. As a consequence of persistently elevated glucose levels, AGEs accumulate in diabetic ulcers, where they trigger chronic inflammation, tissue damage, defective angiogenesis, and re-epithelialization [56,57]. Importantly, the pathological effects of AGEs are mediated by the activation of signalling cascades via the receptor for advanced glycation end products (RAGE), which is a 45 kDa transmembrane receptor of the immunoglobulin superfamily. Thus, therapeutic agents that can reverse AGE formation or inhibit AGE–RAGE signalling may exert benefits on diabetic foot ulcers. One study found that AGEs lead to M1 macrophage polarization (increased CD11c population) and the production of pro-inflammatory cytokines via the up-regulation of autophagy, leading to impaired wound healing [58]. Another study demonstrated that AGE–RAGE signalling causes disruption in macrophage function in vivo, including their phagocytosis ability and M1-to-M2 polarization [59]. The application of an anti-RAGE antibody reversed these effects. Moreover, AGEs may alter the signalling of cytokines and growth factors by disrupting the structure of either the growth factors or their receptors [57], which also contribute to the hostile environment of diabetic wounds. Recently, small molecules that inhibit the AGE–RAGE axis have been shown to be good candidates for alleviating diabetic complications; however, no clinical study has been performed on their application to diabetic wound healing.

### 5.4. Mesenchymal Stem Cell (MSC)-Derived Exosomes

Mesenchymal stem cells (MSCs) have been widely applied in regenerative medicine because of their multipotency. Interestingly, the therapeutic potential of MSCs can be attributed to MSC-derived exosomes (MSC-exosomes) [60], which contain DNA, RNA (including miRNA), lipids, proteins, and metabolites. Despite the potential of naturally derived exosomes, pre-conditioning methods have been developed to enhance the bioactivity or production of MSC-exosomes [61]. In one case, when MSCs were pre-treated with LPS, the exosomes (LPS pre-Exo) harvested in this condition improved diabetic wound healing [62]. Further investigation showed that an miRNA (let-7b) present in the LPS pre-Exo down-regulated TLR4/NF-κB expression while enhancing STAT3/AKT signalling, which resulted in increased M2 macrophages in the exosome-treated wounds. In a separate study, melatonin-pre-treated MSC-derived exosomes (MT-Exo) potently suppressed the pro-inflammatory cytokines IL-1β and TNF-α and promoted the anti-inflammatory factor IL-10 [63]. When applied in vivo, MT-Exo reduced inflammation by increasing M2/M1 polarization and promoted subsequent angiogenesis and collagen synthesis, thus favouring cutaneous wound healing. Compared with stem cell therapy, exosomes derived from MSCs may hold a similar potential with fewer concerns regarding safety and immunocompatibility. Future applications of exosome-based strategies in translational medicine are warranted [21]; however, their cost and large-scale production could be major hurdles that need to be overcome.

### 5.5. MicroRNAs

Several studies have reported that an imbalance in microRNA (miRNA) expression may be a key factor underlying non-healing diabetic wounds [64]. Here, we provide several examples of anti-inflammatory or pro-inflammatory miRNAs and their applications. Keratinocyte-derived miR-132 has been shown to inhibit the expression of pro-inflammatory cytokines in keratinocytes, monocytes, and macrophages by regulating the NF-κB, NOD-like receptor, TLR, and TNFα signalling pathways; in turn, miR-132 was also shown to promote M2 polarization in macrophages [65]. Interestingly, miR-132 down-regulation has been observed in human diabetic wounds compared with normal-healing wounds, whereas a liposome-formulated miR-132 mimetic mixed with Pluronic F-127 gel induced re-epithelialization in an ex vivo human wound model [66]. In addition, miR-146a expression was down-regulated during diabetic wound healing, resulting in increased levels of IRAK1, TRAF6, and NF-kB, which are involved in pro-inflammatory signalling pathways [67]. Notably, treatment of diabetic wounds with MSCs corrected the impaired expression of miR-146a and accelerated the wound-healing process [67]. Importantly, the use of cerium oxide nanoparticles conjugated with miR-146a also enhanced wound closure in db/db mice [68]. In contrast with the roles of miR-132 and miR146a, miR-155 was found to be overexpressed in the skin of patients with diabetes. The inhibition of miR-155 significantly reduced the infiltration of inflammatory cells, whereas it increased the abundance of M2 macrophages [69]. Moreover, miR-155 knockdown restored the expression of fibroblast growth factor 7 (FGF7), which in turn promoted keratinocyte migration and proliferation [70]. Increased vascular remodelling and enhanced re-epithelization also accompanied miR-155 inhibition in a murine diabetic model [70].

In addition to microRNAs, other non-coding RNAs, including long non-coding RNAs (lncRNAs) and circular RNAs (circRNAs), have received widespread attention, as they also exert regulatory functions in diabetic wound healing [64]. For example, the expression of lncRNA GAS5 was increased in diabetic wounds from mouse and human skin [71]. Moreover, overexpression of lncRNA GAS5 up-regulated STAT1 and promoted the M1 polarization of macrophages in vitro, while topical administration of an shRNA against GAS5 in diabetic wounds significantly accelerated the wound-healing process [71]. Nevertheless, additional pre-clinical and translational studies are needed to assess the treatment potential of each of these non-coding RNA molecules because of their inherent complexity and diverse modes of action. In addition, gene-based approaches have been limited by the efficiency and specificity of the delivery systems.

### 5.6. Stem Cell Therapy

Current studies have demonstrated that MSCs promote chronic wound healing by regulating macrophages [72,73,74,75,76]. Briefly, Uchiyama et al. found that the expression of the MEG-E8 glycoprotein was decreased in the granulation tissue of chronic wounds in db/db mice. As MSCs have been reported to produce abundant MFG-E8, the subcutaneous injection of MFG-E8 WT and MFG-E8 knockout (KO) MSCs into the wounds of db/db mice revealed that WT MSCs promote chronic wound healing by down-regulating TNF-α and up-regulating IL-10. Moreover, the infiltration of M2 macrophages was promoted by MFG-E8 [73,74]. However, the manner in which MEG-E8 modulates the secretion of distinct cytokines and their target cells for regulating M1 or M2 macrophages has not been investigated.

Autologous transplantation of bone-marrow-derived MSCs has been shown to promote DFU healing in clinical trials [77,78]. Moreover, PGE2 secreted from umbilical-cord-derived MSCs rescued the dysfunction of endothelial cells and improved angiogenesis via the regulation of M1-to-M2 macrophage polarization in diabetic wounds [76]. Transplantation of placenta-derived MSCs has also been shown to promote diabetic wound healing by decreasing the levels of the TNF-α, IL-6, and IL-1 pro-inflammatory cytokines and inhibiting NF-κB signalling [79].

Adipocyte-derived stem cells (ADSCs) have been attracting attention as an effective therapeutic tool for tissue regeneration. One study demonstrated that ADSC-derived exosomes enhanced diabetic wound healing via the circular RNA Snhg11, which can promote M2 macrophage polarization by regulating the miR-144-3p/HIF-1α/STAT3 signalling pathway [80]. Another study demonstrated that the overexpression of hematopoietic prostaglandin D synthase (HPGDS), which is a cytosolic protein that can convert prostaglandin H2 (PGH_2_) to prostaglandin D2 (PGD_2_), in ADSCs could accelerate chronic wound healing by improving the anti-inflammatory state and promoting M2 macrophage polarization [81].

### 5.7. Antioxidant Therapeutics

An exceedingly high level of oxidative stress and a significant reduction in antioxidant enzyme activity are considered as the key causes of non-healing diabetic wounds. M1 macrophages use their anti-bacterial function as the primary defence against invading pathogens [82]. M1 macrophages trigger the anti-bacterial response through the production of nitric oxide (NO), reactive oxygen species (ROS), interleukin-1 (IL-1), and tumour necrosis factor (TNF), in order to clear and damage pathogens [83]. In addition, ROS may serve as the secondary messenger in TNFα-induced death by inhibiting MAP kinase phosphatases and activating NF-κB activity through the tyrosine phosphorylation of IκBα [84,85]. In contrast to M1 macrophages, M2 macrophage activation is accompanied by reduced ROS and NO generation through the down-regulation of NADPH oxidases 1 and 2 (NOXs) and nitric oxide synthase (iNOS) and/or the up-regulation of arginase, leading to anti-inflammatory, tissue-remodelling, and wound-healing effects [86,87,88].

The oral administration of antioxidants (vitamins C/E and *N*-acetyl cysteine (NAC)) has been applied to the investigation of the regulation of ROS in diabetic wound healing. Oral inoculation of high-concentration vitamins C/E and NAC could accelerate wound healing through antioxidant and anti-inflammatory activity [89,90]. Additional results demonstrated the oral-antioxidant-induced trend toward a decrease in the MIG^+^/CD206^−^ M1 macrophage population, whereas the CD206^+^/MIG^−^ M2 macrophage population was not restored after high-concentration vitamin C/E treatment. These results suggest that oxidative stress plays a major role in diabetic wound-healing impairment and that the oral administration of antioxidants improves healing by modulating inflammation, rather than via M2 macrophage regulation [89].

A recently designed topical antioxidant treatment exhibited therapeutic potential in diabetic wound closure [91]. This ROS-scavenging hydrogel was developed using polyvinyl alcohol (PVA), which has been demonstrated to decrease ROS levels and up-regulate M2 phenotype macrophages around the wound site [91]. Such a device could be loaded with different types of drugs; in this case, the author chose mupirocin to kill bacteria and granulocyte/macrophage colony-stimulating factor (GM-CSF) to accelerate wound closure. However, pre-clinical results are required to further support the efficacy of this type of ROS-scavenging hydrogel.

Plant ingredients have been considered as natural antioxidants and are widely used in wound-healing treatments. Ferulic acid, paeoniflorin, and syringic acid are naturally derived antioxidant phenolic compounds that are often found in fruits and vegetables; they inhibit lipid peroxidation and increase the expression of catalase, superoxide dismutase, glutathione, and nitric oxide, thus improving the healing process of diabetic ulcers [92,93]. However, the role of phenolic compounds in the regulation of macrophages remains unclear. Recently, quercetin, a natural polyphenol, was demonstrated to inhibit M1 polarization, ROS production, and phagocytosis. In contrast, quercetin also enhanced M2 macrophage polarization and endogenous antioxidant expression in vitro [94]. These results suggest that quercetin may act as a potential compound that modulates the M1/M2 transition and may contribute to diabetic would healing. Lupeol, which is a plant-derived flavonoid, remarkably inhibited M1 macrophage polarization (F4/80^+^iNOS^+^) while promoting M2 macrophage polarization (F4/80^+^CD206^+^) in rats with diet-induced metabolic syndrome [95]. Moreover, lupeol accelerated wound healing in streptozotocin-induced hyperglycaemic rats through a decrease in the expression of NF-κB and IL-6 and an increase in IL-10, FGF-2, TGF-β1, HIF-1α, Ho-1, and Sod-2 expression [96]. These findings indicate that lupeol possesses wound-healing potential in hyperglycaemic conditions and may be useful as a treatment for chronic wounds in patients with diabetes.

### 5.8. Topical Application of ON101

ON101 is the first macrophage-regulating agent to be approved for the treatment of diabetic foot ulcers topically in Taiwan. ON101 was formulated using identified, defined fractions of *Plectranthus amboinicus* (PA-F4) and *Centella asiatica* (S1) in a proprietary ratio [97]. The mechanism via which ON101 promotes diabetic wound healing occurs dually through the attenuation of M1 macrophage polarization and the enrichment of M2 macrophages. PA-F4, which is the major active pharmaceutical ingredient (API) of ON101, exerts its anti-inflammation action by suppressing LPS-induced macrophage activation via the down-regulation of NF-κB-mediated NLRP3 inflammasome activation, thus inhibiting the release of IL-1β, IL-18, and IL-6 from macrophages [98]. In addition, administration of ON101 during the polarizing progression of M1 macrophages from either a THP-1 monocyte cell line or monocytes isolated from peripheral blood mononuclear cells (PBMCs) results in the suppression of the expression of genes encoding the M1 markers CD80 and CD86 [97]. The expression of M1-associated cytokines (IL-6, IL-1β, and TNF-α) and chemokines (CXCL1, CXCL9-12, and CCL12) was also suppressed by ON101. Conversely, chemokines including CCL2, IL-4, and CCL3, which have been shown to display M2 macrophage-recruiting abilities [25,53,54,99], were up-regulated upon ON101 treatment, suggesting that M1 macrophage suppression leads to the alleviation of the chronic inflammatory cytokine profile and alters the microenvironment to favour M2 macrophage recruitment [97]. Furthermore, ON101 stimulated the expression of GCSF and CXCL3 from adipocyte progenitor cells (ADPCs), which are a type of skin-resident mesenchymal stem cell. In vitro treatment with GCSF or CXCL3 recombinant proteins demonstrated that GCSF enhanced CD206 and CD163 expression, whereas CXCL3 enhanced CD163 expression in M1 macrophages, suggesting that these two proteins promote the M1-to-M2 transition [97]. A genome-wide screening would be critical to explore the manner in which ON101 deeply alters the behaviour of ADPCs.

Based on its ability to dually modulate both the M1 and M2 macrophage subtypes, ON101 promoted healing efficacy in patients with DFUs in a multicentre randomized clinical trial (NCT01898923) [100]. In that study, 236 eligible patients with DFUs classified as Wagner grade 1 or 2 were randomized to receive either the ON101 cream (*n* = 122) or an absorbent dressing (*n* = 114) for a period as long as 16 weeks. The incidence of complete healing was 74 patients (60.7%) in the ON101 group and 40 patients (35.1%) in the control dressing group (difference, 25.6 percentage points; odds ratio, 2.84; 95% CI, 1.66–4.84; *p* < 0.001). A subgroup analysis of DFU-related risk factors, including a glycated haemoglobin level ≥ 9%, an ulcer area > 5 cm^2^, and DFU duration ≥ 6 months, also demonstrated that ON101 afforded a better healing incidence than the conventional dressing [100]. These findings provided a proof-of-concept with respect to the reshaping of the macrophage subtypes that are critical in the co-ordinated process of healing of complicated dermal wounds clinically.

The above-mentioned macrophage-regulating approaches are listed in Table 1.

## 6. Future Perspectives

DFUs exhibit a growth rate of 9% annually because of the increased incidence of diabetes in the developed world. DFUs are a serious chronic wound associated with a major health care burden and significant morbidities and mortalities. Unfortunately, the current staggeringly high number of DFUs is growing much faster than the emergence of new effective therapies. Multi-disciplinary approaches to maintain blood sugar, infection control, vessel reconstruction, off-loading devices, life-style coaching, and wound management are the key and initial steps to prevent DFU complications and limit associated amputations and morbidities. New research on chronic wound mechanisms, such as macrophage polarization, provides a new horizon for examining chronic wounds and developing new drugs for DFUs. The oral administration of vitamin E and C antioxidants improves wound closure by decreasing the MIG/CD206 M1 macrophage population and related inflammatory cytokines. A ROS-scavenging hydrogel promotes wound closure by reducing ROS levels and up-regulating M2 phenotype macrophages around the wound. Moreover, the application of topical insulin or CCL2 chemokines may reverse high-glucose-induced pro-inflammatory cytokines; promote macrophage migration into wound beds; enhance the efferocytosis of apoptotic neutrophils by macrophages; promote the transition from M1 to M2 polarization; and induce the secretion of VEGF, TGF-β, and EGF. However, high-quality clinical evidence in this context is limited, the therapeutic efficacy in human DFUs is unknown, and drug stability and delivery remain major challenges. Stem-cell-related therapy brings new hope for the management of chronic wounds because of the suppression of the pro-inflammatory cytokines IL-1β, TNF-α, and IL-6; the inhibition of iNOS expression; the stimulation of growth factor secretion; and the promotion of M2 macrophage polarization (F4/80^+^/CD206^+^) and collagen I synthesis to induce wound healing. The high cost and efficacy of these agents remain the major barriers to their clinical application. Macrophage-regulating drugs are a beacon for DFUs, as demonstrated in an MCRT trial. Topical application is easy and convenient for home care. This product showed a clinically and statistically superior therapeutic efficacy regarding the complete healing rate and the time to complete healing compared with the hydrocolloid dressing in the treatment of DFUs. This result is superior to the DFU studies that evaluated platelet-derived growth factor (50% wound healing), bioengineered tissues (30–56% wound healing) [102,103], and NPWT (43–56% wound healing). Further studies in patients with different diabetic ulcer stages and DFUs combined with infection or nephropathy in dialysis are still needed.

## Figures and Tables

**Table 1 pharmaceutics-14-02065-t001:** List of novel macrophage-regulating approaches for DFU treatment.

Approach	Mechanism (Itemized, Related to Macrophages)	Stage (In Vitro/Pre-Clinical/Phase I/II/III)	Advantages/Disadvantages (Itemized)	References
**Type: Antioxidants**				
Oral administration of vitamins E and C (40 and 100 mg/kg b.w.)	Improved wound closure by decreasing the MIG^+^/CD206^−^ M1 macrophage population and related inflammatory cytokines	Pre-clinical diabetes was induced by monohydrate alloxan i.v. administration (70 mg/kg b.w.)	**Advantages:** Known safety profile Low cost M1 inhibition **Disadvantages:** No clinical evidence No M2 regulation evidence	[89]
ROS-scavenging hydrogel	Promoted wound closure by decreasing ROS levels and up-regulating M2-phenotype macrophages around the wound	Diabetic mouse model induced by STZ	**Advantages:** Novel approach Local administration **Disadvantage:** No clinical evidence	[91]
**Type: Plant extracts**				
Quercetin	Inhibited M1 polarization by regulating ROS production and phagocytosis; enhanced M2 macrophage polarization and endogenous antioxidant expression	In vitro, the M2 macrophage marker CD206 could be induced by quercetin in RAW264.1 macrophage cells	**Advantage:** Dual modulation of both M1 and M2 macrophages **Disadvantages:** No pre-clinical proof-of-concept based on animal experiments No clinical evidence	[94]
Lupeol	Promoted M1 macrophage polarization (F4/80^+^/iNOS^+^); elevated M2 macrophage polarization (F4/80^+^/CD206^+^)	Diet-induced metabolic syndrome in rats	**Advantage:** Dual modulation of both M1 and M2 macrophages **Disadvantages:** No proof-of-concept based on pre-clinical animal experiments No clinical evidence	[95]
ON101	Directly suppresses M1 macrophage polarization and M1-mediated pro-inflammatory cytokine secretion (IL-1β, TNF-α, and IL-6); suppressed the NLRP3-mediated inflammasome activation through the NF-κB pathway in pro-inflammatory macrophages; stimulated the expression of the genes encoding CXCL3 and GCSF from ADPCs, thus promoting the M1-to-M2 transition	In vitro: directly suppressed M1 markers and M1-associated cytokine secretion Animal model: Promoted diabetic wound healing and enriched M2 macrophages in diabetic wound mouse models (db/db mouse model and HFB mouse model) A phase III multicentre randomized clinical trial (NCT01898923) demonstrated that ON101 exhibited better healing efficacy than the absorbent dressing alone in the treatment of DFUs	**Advantages:** Dual modulation of both M1 and M2 macrophages through independent pathways Clinical proof-of-concept on the acceleration of diabetic foot ulcers with high safety Approved by the Taiwan FDA and Pharmaceutical Administration Bureau of Macao SAR Government, China for the treatment of DFUs in 2021 **Disadvantages:** Complicated composition of the drug product	[97,98,100]
**Type: Growth factors and cytokines**				
Topical insulin administration (injectable solution, cream, dressing, and gel)	1. Reversed high-glucose-induced pro-inflammatory cytokines 2. Promoted macrophage migration into wound beds 3. Enhanced the efferocytosis of apoptotic neutrophils by macrophages 4. Promoted the transition from M1 to M2 polarization	Phase I/II	**Advantages:** Low cost Less safety concerns than systemic administration Positive clinical results **Disadvantages:** High-quality clinical evidence is limited Stable and sustained drug-delivery methods are needed	[45,50,51,52]
CCL2	1. Induced macrophage infiltration into wound sites 2. Promoted the pro-healing function of macrophages through the secretion of VEGF, TGF-β, and EGF	Pre-clinical diabetic mice induced by STZ and db/db mice	**Advantages:** Effective treatment in animal models Clear mechanisms **Disadvantages:** Therapeutic efficacy in human DFUs is unknown Stability and drug delivery are major challenges	[27,53,54,101]
**Type: Exosomes**				
Exosomes isolated from LPS-treated MSCs	1. Enriched with the let-7b miRNA, which regulates macrophages through TLR4/NF-κB and AKR signalling 2. Increased M2 macrophages in exosome-treated wounds	Pre-clinical diabetic rats induced by STZ	**Advantages:** Attractive alternative to stem cell therapy Relatively safe compared with cell therapy **Disadvantages:** No clinical evidence Could be very expensive Difficult for large-scale production	[62]
Exosomes isolated from melatonin-conditioned MSCs	1. Suppressed the pro-inflammatory cytokines IL-1β and TNF-α while promoting IL-10 expression in vitro 2. Increased the M2/M1 ratio in the wounds	Sprague Dawley (SD) rats treated with STZ		[63]
**Type: MicroRNAs and long non-coding RNAs**				
Liposome-formulated miR-132 mimics mixed with gels	1. Inhibited the expression of pro-inflammatory cytokines in keratinocytes, monocytes, and macrophages by regulating NF-κB, NOD-like receptors, TLRs, and TNF-α 2. Promoted the M2 polarization of macrophages	Pre-clinical db/db mice and human ex vivo wound model	**Advantages:** Novel approach Regulates multiple targets **Disadvantages:** Delivery efficiency Off-target effects No clinical evidence Complicated biology	[66]
shRNA against LncRNA GAS5	1. lncRNA GAS5 up-regulated STAT1 levels and promoted the M1 polarization of RAW macrophages 2. Knockdown of lncRNA GAS5 accelerated wound healing with decreased IL-6 and TNF-α levels	In vitro RAW264.7 cells In vivo db/db mice		[71]
**Type: Stem cell therapy**				
Subcutaneous injection of mouse bone-marrow-derived MSCs	Decreased TNFα expression and increased Il-10 expression via MSC-derived MFG-E8; increased the number of CD68^+^ /arginase-1^+^ M2 macrophages via MSCs-derived MFG-E8	db/db diabetic mouse model	**Advantages:** Effective treatment in diabetic animal model Improves diabetic wound healing by multiple functions **Disadvantages:** Clinical potential of MSC-derived MFG-E8 in diabetic wounds is unknown High cost	[73,74]
Subcutaneous application of (1) human-placenta-derived MSCs and (2) infant umbilical-cord-derived MSCs	Suppressed the pro-inflammatory cytokines TNF-α and IL-6 and increased the anti-inflammatory cytokine IL-10 Suppressed the pro-inflammatory cytokines IL-1β, TNF-α, and IL-6 and increased the percentage of CD206^+^/arginase-1^+^ M2 macrophages	1. Diabetic Goto–Kakizaki (GK) rats 2. STZ-induced diabetic mouse model	**Advantages:** Regulates multiple targets Placenta- and umbilical-cord-derived MSCs have been used in clinical treatment of DFU and have shown promising results. **Disadvantage:** High cost	[76,79]
Adipocyte-derived MSCs	Suppressed the pro-inflammatory cytokines IL-1β, TNF-α, and IL-6; inhibited iNOS expression; promoted M2 macrophage polarization (F4/80^+^/CD206^+^) and collagen I synthesis	High-fat diet plus STZ-induced type 2 diabetic mouse model (late-stage type 2 diabetes-like model)	**Advantage:** Clear mechanisms **Disadvantages:** High cost Clinical importance of HPGDS in diabetic wounds needs to be further elucidated How the decreased HPGDS leads to delayed diabetic wound healing needs to be further addressed	[81]

Abbreviations: ADPCs, adipocyte progenitor cells; HFB, high-fat diet; DFU, diabetic foot ulcer; TFDA, Taiwan Food and Drug Administration.

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
