# Peer review of "New Horizons of Macrophage Immunomodulation in the Healing of Diabetic Foot Ulcers"

_pharmaceutics, 2022, doi:10.3390/pharmaceutics14102065_

Round 1

Reviewer 1 Report

The article reviews diabetic foot ulcers (DFUs) in innovative approaches. It is a significant contribution to the field because the wound chronicity of DFUs still is a challenge. The article is comprehensively described, starting from the mechanisms of macrophage actions in wound healing until their dysregulation in DFUs, thus giving scientific support to the new approaches regarding the advantages, disadvantages, and limitations of macrophage immunomodulation.  

The article is valuable for scientists looking for a succinct overview of the subject without losing scientific rigor, as it is supported by the relevant bibliography.  

Authored by people involved in biotechnology companies, the article describes novel developments and emphasizes their new topical product ON101, approved by regulatory agencies in China, in terms of effectiveness and ease of application.

Finally, the conclusions are pertinent and pointed out for further studies.

In this context, this reviewer recommends publishing the article in present form. 

Author Response

Thank you for the comments. 

Reviewer 2 Report

The manuscript titled "New Horizons of Macrophage Immunomodulation in the Healing of Diabetic Foot Ulcers" Ching-Wen Lin et al. (2022) reviewed the strategies for diabetic foot ulcers with macrophage immunomodulation. This manuscript is well organized and written, so I have minor comments.

1. Add correspondence information in page 1

2. It would be great if the authors add some references about the material (scaffold) which can change polarization of macrophages.

 1) https://doi.org/10.1016/j.apsusc.2012.03.078

 2) https://doi.org/10.1002/adfm.201909331 

 3) 10.1039/C5TB01605C

 4) https://doi.org/10.1002/adma.202004172

 5) https://doi.org/10.1016/j.jiec.2018.06.010

Author Response

We value your comments and have itemized our responses as below:

  1. Add correspondence information in page 1

Response: We have added the correspondence information in page 1 of the revised manuscript.

  1. It would be great if the authors add some references about thematerial (scaffold) which can change polarization of macrophages.

1) https://doi.org/10.1016/j.apsusc.2012.03.078

2) https://doi.org/10.1002/adfm.201909331

3) 10.1039/C5TB01605C

4) https://doi.org/10.1002/adma.2020041721.

Response: Thank you for the suggestions. We have added one sentence to describe the innovative materials on regulating macrophages in the revised page 3 with yellow labels as follows:

 ï¼‚Of note, numerous innovative biomaterials were created to provide suitable environments for recruiting or polarizing the distinct types of macrophages[16,17], which have been implicated in inflammation-associated conditions and tissue regeneration (as reviewed in[18-20])."
